# Hydrolysis, Biodegradation and Ion Sorption in Binary Biocomposites of Chitosan with Polyesters: Polylactide and Poly(3-Hydroxybutyrate)

**DOI:** 10.3390/polym15030645

**Published:** 2023-01-27

**Authors:** Svetlana Rogovina, Lubov Zhorina, Anastasia Yakhina, Alexey Shapagin, Alexey Iordanskii, Alexander Berlin

**Affiliations:** 1N. N. Semenov Federal Research Center for Chemical Physics Academy of Science, 119991 Moscow, Russia; 2Frumkin Institute of Physics Chemistry and Electrochemistry, Russian Academy of Science, 119071 Moscow, Russia

**Keywords:** polylactide, poly(3-hydroxybutyrate), chitosan, blend composite films, hydrolysis, water sorption, metal sorption

## Abstract

The film binary composites polylactide (PLA)–chitosan and poly(3-hydroxybutyrate) (PHB)–chitosan have been fabricated and their functional characteristics, such as hydrolysis resistance, biodegradation in soil, and ion sorption behavior have been explored. It was established that hydrolysis temperature and acidity of solutions are differently affected by the weight loss of these two systems. Thus, in the HCl aqueous solutions, the stability of the PHB-chitosan composites is higher than the stability of the PLA-chitosan one, while the opposite situation was observed for biodegradation in soil. The sorption capacity of both composites to Fe^3+^ ions was investigated and it was shown that, for PHB-chitosan composites, the sorption is higher than for PLA-chitosan. It was established that kinetics of sorption obeys the pseudo-first-order equation and limiting values of sorption correspond to Henry’s Law formalism. By scanning electron microscopy (SEM), the comparative investigation of initial films and films containing sorbed ions was made and the change of films surface after Fe^3+^ sorption is demonstrated. The findings presented could open a new horizon in the implementation of novel functional biodegradable composites.

## 1. Introduction

Currently, the design of eco-friendly biodegradable polymeric materials, being capable of decomposing in the environment, is a key requirement for the numerous materials exploited in different fields of human activity, namely in biomedicine, packaging, for environment protection as absorbents, and others [1,2,3]. The use of synthetic polymers in food packaging [4] is one of the main causes of pollution in aquatic environments, including oceans, rivers, lakes, and agricultural irrigated lands [5]. At the same time, the presence of toxic ion metals harmful for human health in aqueous media requires the creation of new and efficient biodegradable sorbents based on natural polymers, which can decay after the end of their service life into the harmless substances [6].

To fulfill these requirements, the composite sorbents have been elaborated on the basis of biodegradable polyesters polylactide (PLA) [7,8] and poly(3-hydroxybutyrate) (PHB) [9] as the bio-based polymers synthesized from natural raw materials by chemical and microbiological methods, respectively, and natural polysaccharide chitosan [10]. Both polyesters have the relevant mechanical characteristics close to those of synthetic polymers with high hydrophobicity [11], while chitosan, as a highly hydrophilic biopolymer containing the amino groups, is a high effective sorbent for heavy metals [12]. However, its significant drawback is the swelling in aqueous media and has poor mechanical properties [13]. The development of the biodegradable binary composites based on these polyesters and chitosan allows one to combine mechanical characteristics of PLA and PHB with the high sorption capacity of chitosan. During exploitation, these composition materials are exposed to action of such aggressive factors as hydrolysis, oxidation, ozonolysis and also enzymatic biodegradation via numerous microorganisms [14,15,16,17,18].

In this connection, it is very important to separately evaluate the action of either factor on the complex of properties of investigated systems for their successful use as biodegradable sorbents of metal ions from aqueous media.

It is well known that, in the hydrophobic polyesters, the water sorption capacity is extremely poor and amounts to about a percent or even less [19]. For example, the maximal water capacity in PLA exposed in aqueous medium during a month is 1.3% [20]. By contrast, the hydrophilic chitosan is characterized by high water sorption values [21].

Evaluation of environmental disintegration of composite materials based on PLA and PHB is extremely important, since their ability to biodegrade is one of the key motivations for the use of these polyesters in the creation of green composites. It should be noted that the biodegradation rates of PLA and PHB differ essentially [22]. As shown in [23], PHB loses 14.2% of the initial mass within six months, while the mass of PLA samples remains practically unchanged over the same period of time. This fact can be explained by different mechanisms of PLA and PHB biodegradation. If the biodegradation of PHB in the soil occurs under the action of microorganisms, the biodegradation of PLA initially occurs hydrolytically, and then it proceeds enzymatically in accordance with the two-stage mechanism [24]. The non-enzymatic hydrolysis of PLA begins with the sorption of water, leading to the breaking of ester bonds and the formation of low molecular weight oligomers and the monomer of lactic acid [25]. It should be noted, however, that at elevated temperatures (more than 50 °C), the weight loss of PLA in the compost within the four weeks can reach 45 wt. % [26]. This is explained by PLA hydrolysis proceeding readily above its glass transition temperature (55–62 °C) [27]. Previously, in [28,29], we studied the features of hydrolysis and ozonolysis of PLA, PHB, and ternary compositions based on PLA, PHB, and chitosan, as well as their sorption properties with respect to metal ions [30]. The elaborated PLA–PHB binary compositions were successfully used for oil sorption from aqueous media [31]. The simultaneous use in the composition of two polyesters PLA and PHB performing a reinforcing function improved their mechanical strength, but made it difficult to estimate the effect of each of the polymers used in the process. The purpose of this work was to study the features of acid hydrolysis, soil biodegradation, and iron ion sorption of the binary polymeric composites’ films, namely PLA-chitosan and PHB-chitosan as the biodegradable and eco-friendly materials potentially designated for metal sorption and packaging.

The novelty of the obtained data is connected to the establishment of different influences of PLA and PHB on the complex of properties of the investigated binary film composites polyester–chitosan.

## 2. Materials and Methods

### 2.1. Materials

PLA 4043D from Nature Works (Minnetonka, MN, USA) as pellets with diameter of 3 mm (M_w_ = 2.2 × 10^5^, M_n_ = 1.65 × 10^5^, T_m_= 155 °C, polydespersity index D = M_w_/ M_n_ = 1.35, transparency 2.1%); PHB (Biomer, Kreilling, Germany) (M_w_ = 2.05 × 10^5^, T_m_ = 175 °C), chitosan produced by Bioprogress (Shchelkovo, Moscow region, Russia, M_w =_ 4.4 × 10^5^, degree of deacetylation 0.87), and anhydrous ferric chloride (Fluka Chemie, Buchs, Switzerland) were used.

### 2.2. Production of Compositions

Films were prepared by mixing PLA and PHB solutions in chloroform, in which chitosan in the form of a powder under mechanical stirring was introduced. The resulting films with a thickness of 0.2–0.3 mm were dried at room temperature.

### 2.3. Hydrolytic Degradation 

Hydrolysis of the obtained films was proceeded in 0.005, 0.1 and 0.2 mol/L aqueous HCl solutions at temperatures 25, 40 and 70 °C. The samples were removed from HCl solution at certain intervals and placed in an oven for 2 h at a temperature of 90 °C after which they were weighed (measurement error d = 0.1 mg).

### 2.4. Biodegradability

The biodegradability of the obtained films was studied by modeling the processes occurring under the natural condition (ASTM D5988-12). The samples were placed in soil (pH = 7.5) for several weeks with the following measure of its weight loss at certain time intervals.

### 2.5. X-ray Fluorescence Analysis

For the investigation of iron ions sorption by composition films, they were placed in aqueous solutions of FeCl_3_ of various concentrations and kept. The percentage content of sorbed iron ions in the film compositions was determined by method of X-ray fluorescence analysis on an ARL PERFORM’X X-ray Fluorescence spectrometer (Thermo Fisher Scientific, Waltham, MA, USA). Registration of spectra and all further manipulations with them were performed using the SIALMO.UQ method.

### 2.6. Scanning Electron Microscopy (SEM)

The morphology of the composite films PLA-chitosan and PHB-chitosan before and after the sorption of iron ions was studied by a method of scanning electron microscopy (SEM) using a Philips SEM-500 scanning electron microscope (The Netherlands) in the secondary electron mode at an accelerating voltage of 15 keV. Sample preparation consisted of thermal vacuum deposition of carbon on the film surface using a VUP-5 vacuum universal post (Russia). To obtain a continuous conductive carbon coating, the samples were rotated in the same plane during thermal spraying.

## 3. Results and Discussion

### 3.1. The Comparison of Hydrolysis Behavior for Binary Systems PLA-Chitosan and PHB-Chitosan

The durability and efficiency in using the binary composites based on the biopolyesters and chitosan as the toxic metal sorbents from aqueous media are largely determined by their ability for hydrolytic or enzymatic degradation. In this work, in the temperature range 25–70 °C, the hydrolysis of binary compositions PLA-chitosan and PHB-chitosan in aqueous hydrochloride acid solutions with various concentrations was studied. The results obtained are presented in Figure 1, Figure 2 and Figure 3.

As can be seen from the above Figures, most of the kinetic curves have two characteristic ranges. In the initial time there is a very sharp weight loss of films that takes place, after that, with the exit on the plateau or with a very slow change, weight loss is observed. The exceptions belong to the PLA-chitosan specimens exposed in extremely aggressive media, namely 70 °C; and the highest HCL concentrations. For such systems, the second kinetic stage proceeds with a remarkable rate that could be described by the linear dependence.

Based on the previous papers devoted to the study of diffusional kinetics of controlled drug release from homopolymers PHB and PLA [32,33,34], it is possible to propose that the initial fast stage is governed by the diffusional delivery of low molecular fraction of the polymeric components from the chitosan hydrogel matrix. The second stage is probably related to the destruction of the polyesters, PHB and PLA, that under given conditions occurs extremely slowly due to their essential crystallinity. At the given range of temperatures, all kinetic curves for PLA are located slightly higher than for PHB that again confirms somewhat higher resistance to hydrolysis of the member of the polyhydroxyalkanoates’ family. It is quite clear that the hydrolysis of PLA-chitosan composition proceeds more intensively than the PHB-chitosan one, which is explained by the minor difference in the chemical structure of both polyesters, namely due to the diversity of molecular hydrophilic/hydrophobic balance in the PLA and PHB chains [35].

It should be noted that at the lowest temperature 25 °C; for the compositions PLA-chitosan and PHB-chitosan an increase in the acidity of solutions leads to the significant increases in the weight loss, but the form of the kinetic curves besides the system PLA-chitosan in 0.2 mol/L HCl solution remains the same. However, under the severest conditions at 70 °C, the weight loss kinetics curve of PLA-chitosan composition is remarkably changed in comparison with the PHB chitosan one.

### 3.2. Sorption of Fe^3+^ Ions by the Binary Systems PHB-Chitosan and PLA-Chitosan

The binary compositions based on biodegradable polyesters and chitosan are of great interest for their use as metal absorbents from wastewater. PLA and PHB polyesters have good mechanical properties, while chitosan, which easily swells in aqueous media, binds metal ions well. The study of electrolyte sorption capacity for the binary composite films is of great interest since such films can be easily utilized after the end of their service life without environmental violation. Recently, to estimate the feasibility of PLA–PHB–chitosan as the novel sorbent, we evaluated the sorption constants of iron and chromium ions by the above-mentioned ternary system [36]. In the present publication, to continue consideration of the ionic pollutant sorption behavior, by somewhat simplified binary systems PLA-chitosan and PHB-chitosan for which the relevant combination of good mechanical characteristics provided by the polyesters and a high ion capacity via chitosan is successfully implemented. Besides, the ion sorption comparison for these two binary composites enables the authors to reveal the role of interactions between the ester groups for the polyesters (PLA and PHB) and amino groups for chitosan. Additionally, and more important, the exploration of the binary sorbents enables the experts to estimate the difference in pollutant sorption behavior and compare the impact of both polyesters upon the chitosan capacity of Fe^3+^ ions.

The effectual recovery of ionic pollutants from the aqueous environment, primarily we say about the ions Fe^3+^, is determined not only by the chemical structure, the morphology, and the pore pattern of biopolymer sorbent, but such key characteristics as thermodynamic affinity of the electrolyte generating the ions to functional groups of the sorbent, as well as pseudo-chemical kinetics and diffusion of the ionic pollutant [37,38]. Both kinetic processes enable the specialists to provide the complete decontamination of aqueous polluted media and to evaluate the optimal conditions for the polymer sorbent operation [39].

#### 3.2.1. Kinetic Aspect of Fe^3+^ Sorption

The typical kinetic curves, expressing the dependences of Fe^3+^ concentration (wt. %) in the bulk compositions PLA-chitosan and PHB-chitosan (sorption, Cp) on the time of exposition (t) are presented in Figure 4a,b correspondingly. As is shown in these Figures, all the curves have well-defined limits, the values of which are determined by the concentration of the electrolyte, FeCl_3_, in the aqueous volume (Cv). Within the interval 0.002–0.008 mol/L, with the increase in Cv, the limited values of sorption Cp_∞_ are increased as well. Last finding corresponds to the positive increment of the functional dependence Cp = f(Cv) featured for most of the sorption isotherms, such as Langmuir, BET, GAB and others [40].

The comparison between two families of the kinetic curves for PLA-chitosan and PHB-chitosan systems showed that the limiting sorption concentration for the latter considerably exceeds the same characteristic of the former. In other words, the replacement of PLA by PHB in the chitosan-polyester composite leads to the increase in Fe^3+^ more than 100 wt. %. A reasonable cause for the observed effect may be the reduction in the ester groups interaction with amino groups, and as a consequence, the disengagement of the chitosan functional groups for the interaction with Fe^3+^ ions that essentially enhances their sorption. More specifically, the analysis of this assumption will be carried out in our next submission.

In the pseudo-first-order kinetic model of sorption, the rate of ions occupancy in the biopolymer is proportional to the amount of free active centers being available for ion immobilization [39,40]. The simplified expression of this fundamental statement in the differential form is:dq/dt = k_a_(q_∞_ − q),(1)
where q is the amount of the active centers of ion sorption in the biopolymer, q_∞_ is the constant characteristic corresponding to the limit value of occupied center at t → ∞, t is current time of sorption and k_a_ is the effective constant of sorption.

The integration of Equation (1) leads readily to its own sorption kinetic equation presented in the ion concentration scale:Cp = Cp_∞_[1 − exp(−kt)],(2)
where Cp and Cp_∞_ are correspondingly the current and limited values of ion sorption measured at a given moment t and the limited value at t → ∞ respectively; k is the effective kinetic constant of the first order differential Equation (1).

The most of ion uptake systems consisted with Equations (1) and (2) are treated in the semi-logarithmic scale to present the sorption results in the suitable form of the constants evaluation:ln[1 − (Cp/Cp_∞_)] = −kt,(3)

The linearity of Fe^3+^ ions sorption data in the coordinates ln[1-(Cp/Cp_∞_)]~t is shown in Figure 5a,b. The presented lines demonstrate the good conformity between modeling and experimental representations with higher statistical correlation coefficients. All the values of parameters for sorption Equation (3) as well as the corresponding statistical treatment of kinetic curves are presented in Table 1. The statistical coefficients (COP and Pierson coefficients calculated via “Origin2018”^C^ Origin 2018 SR1 Build 9.5.1.195 software) bear evidence to the good correlation of Fe^3+^ sorption modeling with the kinetic experimental data obtained by X-ray fluorescence analysis. It is worth noting that the ion sorption kinetic constant is decreased with the volume electrolyte concentration increment.

The analogous trend was observed in our previous work [30]. It is well known that the process of sorption from aqueous medium is accompanied by transport phenomena that under determined conditions could influence the rate of contaminant recovery [41,42].

For the composites, the growth in the Fe^3+^ ion content could negatively affect the ion mobility in a given polymer matrix and hence extend the effective time of ion achievement to the active sites of sorption. For chitosan, these sites comprise predominantly –NH_2_ groups and to a lesser extend –OH groups. The affinity (sorption thermodynamic impact) of ester entities belonging to PHB and PLA is essentially lower and ion sorption capacity by the polyesters can be neglected as compared to chitosan. This kinetic effect could be probably related to the concentration impact of ion diffusion in electrolyte solution (exterior diffusion) and in the biopolymer matrix (interior diffusion). We have already noted above that the process of sorption from aqueous medium is accompanied by transport phenomena that under determined conditions could influence the rate of contaminant recovery. Regularly, the increase in the concentration of electrolyte (Cv) results in the growth of its content in the biopolymer (Cp), namely in the hydrogel of chitosan as the main hydrophilic component of the composition accumulating Fe^3+^ ions. It is well known that the growth of the ion concentration leads to the enhancement of electrostatic interactions among the mobile ions and hence to the ion diffusivity decrease in a polymer matrix [43]. Consequently, owing to the diffusional hindrance, the time of iron ion access to the functional group of chitosan rises and enlarges the total time of the sorption and decreases the effective sorption constant (k).

#### 3.2.2. Ultimate Fe^3+^ Sorption Capacity for Binary Composites: PLA-chitosan and PHB-chitosan and Henry’s Law Model Description

The model of Fe^3+^ sorption by chitosan reinforced with PLA or PHB based on the reaction of the pseudo first order suggests the ions’ immobilization on functional groups of chitosan (deacetylated -NH_2_ functions). The ultimate values of Fe^3+^ sorption (Cp_∞_) reflect the chitosan state when only part of its active center is occupied by the immobilized ions.

In this case, the linearity of the Cp_∞_ dependence on Cv, (Figure 6) demonstrates the sorption progress in accordance with a Henry’s Law formalism as the initial stage of the Langmuir isotherm [44]:Cp = K_H_ · Cv(4)

The simplified character of the isotherms for both compositions testifies to the lack of interaction among ionogenic groups of the biopolymers. Regularly, such interaction should distort the form of linearity as it occurs in Langmuir or BET modeling. In the interval of FeCl_3_ concentrations spanning the values from 2 × 10^−3^ to 8 × 10^−3^ mol/L, i.e., in the area of diluted concentrations, the ions diffuse to the sorption centers of chitosan independently without interactions with each other. In such a situation, the ion sorption should proceed independently as well, when an amino group does not cooperate with the same adjacent group in the chitosan matrix.

Besides, in the given concentration range, the number of occupied active sites for sorption under saturation (q_∞_ and the corresponding values Cp_∞_) are essentially far from the maximal value of the active sites in chitosan. It could also indicate that the polyesters’ molecules of PHB and PLA partly screen the -NH_2_ groups of chitosan and prevent intermolecular interactions among others. In given situation, we could suggest that the transport of ions is sensitive to their concentration, while the sorption process and the ion interaction with functional groups does not depend on the ion content.

### 3.3. Biodegradation of the Composites PLA-Chitosan and PHB-Chitosan

When studying the destruction of samples which were buried in the soil, it was found that, after exposure of the initial compositions for a month, the mass of the PHB-chitosan composition increased by 36.6 wt. % and the PLA-chitosan composition by 19.8 wt. % accordingly. The observed effect can be associated both with the ability of chitosan to absorb water from the environment and with the growth of microorganisms on the surface of the films. Since, as mentioned above, PHB is more susceptible to the action of microorganisms, it has the largest weight gain. At the same time, for both compositions containing absorbed iron ions, a decrease in the mass of samples by approximately 26 wt. % within a month takes place, that can be explained by the catalytic effect of iron ions. On Figure 7 and Figure 8 the photographs of double films PLA-chitosan and PHB-chitosan as well as photographs of these films, containing sorbet iron ions and after exposition in soil during different times are shown.

The sorption of iron ions leads to a change in the color of the samples from white to red. In addition, with increasing exposure time, their shape also changes. It should be noted that, with long-term exposure (12 weeks), in contrast to the partial destruction of the films based on PLA, the films containing PHB are completely destroyed (Figure 8), which confirms the more intensive process of biodegradation in soil for the composition containing PHB compared to films containing PLA. These data correlate with the features of degradation of compositions based on PLA and PHB described above. At the same time, it can be seen that films that do not contain iron ions are destroyed more easily than films containing iron, which is explained by hardening in the presence of metal ions.

### 3.4. Morphology of Binary Systems PLA-Chitosan and PHB-Chitosan

The morphology of the binary compositions was investigated by the SEM method. In Figure 9a and Figure 10a, the photomicrographs of cross-sections of the films for PLA-chitosan and PHB-chitosan are correspondingly given. As is seen, the freeze surfaces of cross-sections of PLA and PHB are slightly different. For PLA, the surface of the fracture is more of a relief, while the surface of PHB prepared at the same conditions is smoother.

Due to the rather poor compatibility between polyesters and chitosan, two phases are observed. The phase separation is more reliably seen on the polymer specimens containing sorbed ions Fe^3+^ (Figure 9b and Figure 10b). The samples exposition in the aqueous medium of FeCl_3_ leads to chitosan structureless and the formation of areas with the fine pores. In contrast to PLA and PHB, swelling of chitosan under water action forms the smoothed fields situated between the non-swelling polyester entities. The latter should act as cross-linking agents preventing disintegration and dissolution of the chitosan phase.

## 4. Conclusions

The explored binary composites based on chitosan combined with the biodegradable polyesters PLA and PHB have been studied as the novel functional materials being capable to perform in aggressive media, namely aqueous environment and soil. Acid hydrolysis, biodegradation in soil and Fe^3+^ion sorption by PLA-chitosan and PHB-chitosan composites have been studied and obtained results have been compared to understand the merits and drawbacks of the investigated systems.

It was revealed that:In the temperature interval 25–70 °C in the HCl solutions (0.005–0.2) mol/L the PHB-chitosan composite stability exceeds the analogous one for PLA-chitosan composite, while the contrary situation is observed for soil biodegradation;Under studying of the composites biodegradability in soil it was found that the films based on PHB degrade in a greater extent than films based on PLA, that is an additional confirmation of their different biodegradation mechanism;The sorption capacity of both composites to Fe^3+^ions was investigated and it was shown that the kinetics of sorption obeys the pseudo-first order equation and the limiting values of sorption correspond to Henry’s Law formalism. The replacement of PLA by PHB in the chitosan-polyether composites leads to the increase in Fe^3+^ sorption more than on 100 wt. %.

Thus, on the basis of the obtained results, it may be concluded that the PHB-chitosan composites are more stable to the acid hydrolysis, they demonstrate better degradation in soil (may be slightly utilized after end of service life) and have a higher capacity to iron ions sorption than PLA-chitosan composites. In other words, the PHB-chitosan films are more suitable systems for Fe^3^ ions sorption than PLA-chitosan ones.

Both investigated binary film composites on the base of studied biodegradable polyesters and chitosan can be considered as perspective eco-friendly biodegradable materials for use in the packing industry as well for sorbents production. Based on the obtained data, the future research should be dedicated to study thermal stability of binary composites PLA-chitosan and PHB-chitosan as well as their stability to UF action that should allow one to extend the possible areas of their application.

## Figures and Tables

**Figure 1 polymers-15-00645-f001:**
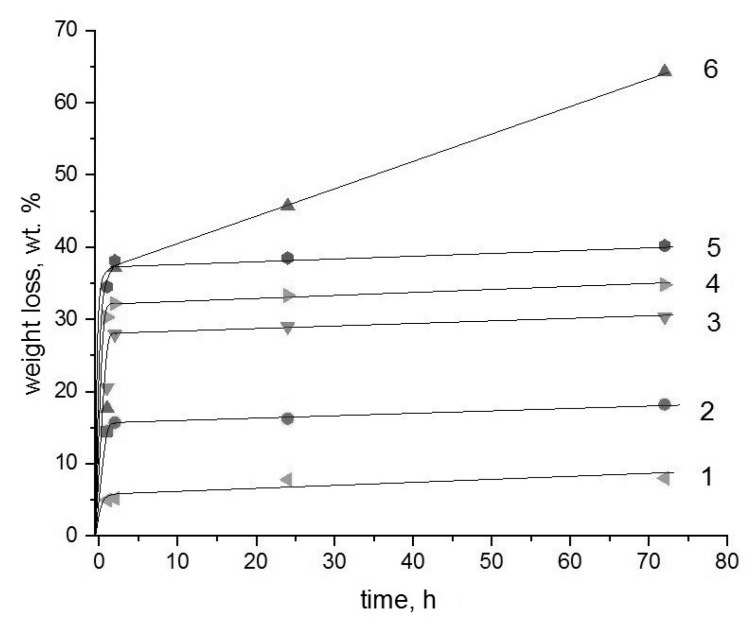
Weight loss kinetics for the binary composite films in the HCl solutions at 25 °C;. The numbers show: Solution concentrations are 0.005 (1), 0.1 (3), 0.2 (5) mol/L for a PHB-chitosan system and 0.005 (2), 0.1 (4), 0.2 (6) mol/L for a PLA-chitosan system.

**Figure 2 polymers-15-00645-f002:**
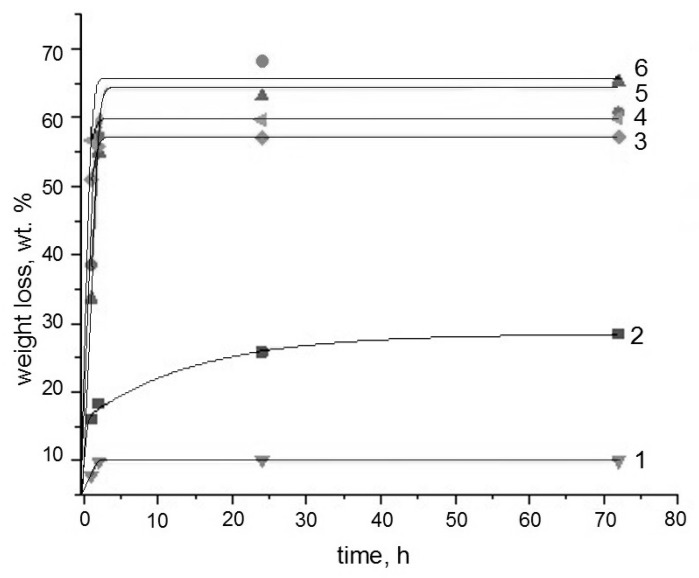
Weight loss kinetics for the binary composite films in the HCl solutions at 40 °C;. The numbers show: Solution concentrations are 0.005 (1), 0.1 (3), 0.2 (5) mol/L for a PHB-chitosan system and 0.005 (2), 0.1 (4), 0.2 (6) mol/L for a PLA-chitosan system.

**Figure 3 polymers-15-00645-f003:**
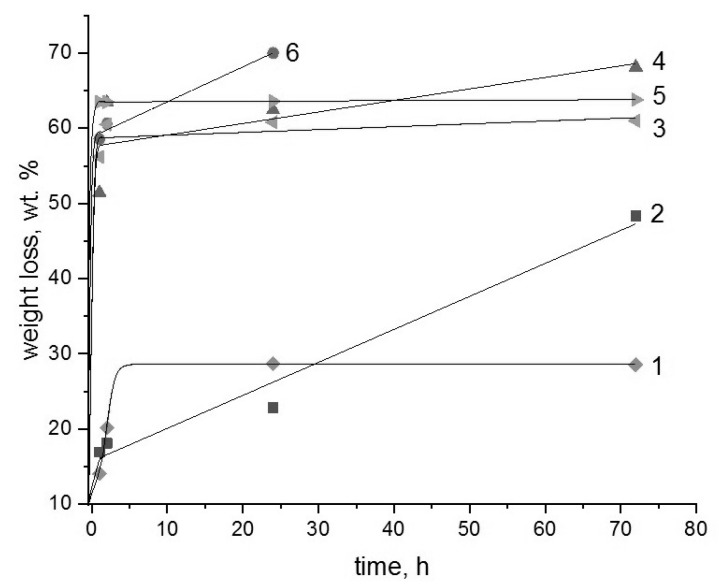
Weight loss kinetics for the binary composite films in the HCl solutions at 70 °C;. The numbers show: Solution concentrations are 0.005 (1), 0.1 (3), 0.2 (5) mol/L for a PHB-chitosan system and 0.005 (2), 0.1 (4), 0.2 (6) mol/L for a PLA-chitosan system.

**Figure 4 polymers-15-00645-f004:**
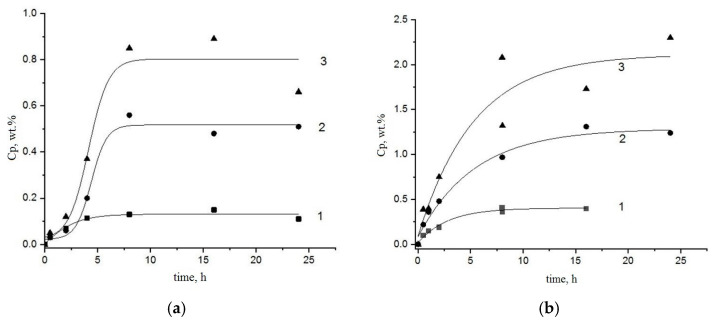
Kinetic curves of Fe^3+^ sorption by PLA-chitosan (**a**) and PHB-chitosan (**b**) films (50:50) wt. % from FeCl_3_ aquatic solutions with different concentrations 0.002 (1), 0.005 (2), 0.008 (3) mol/L.

**Figure 5 polymers-15-00645-f005:**
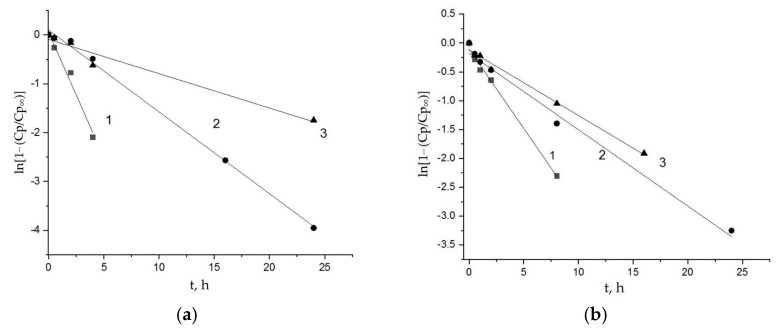
Kinetic curves of Fe^3+^ ions sorption by PLA-chitosan (**a**) and PHB-chitosan (**b**) in semilogarithmic coordinates. Sorption was performed from FeCl_3_ aquatic solutions with different concentrations 0.002 (1), 0.005 (2), 0.008 (3) mol/L.

**Figure 6 polymers-15-00645-f006:**
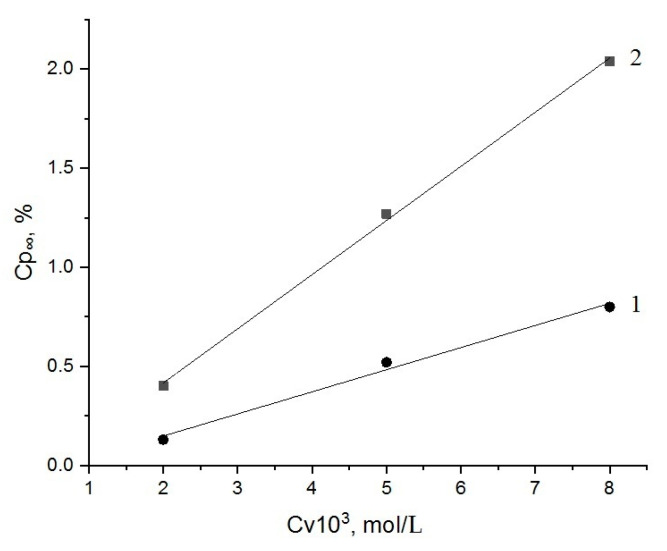
Ultimate Fe^3+^ sorption capacity as the function of concentration in aqueous FeCl_3_ solutions with different molar contents (Cv) for PLA (1) and PHB (2).

**Figure 7 polymers-15-00645-f007:**
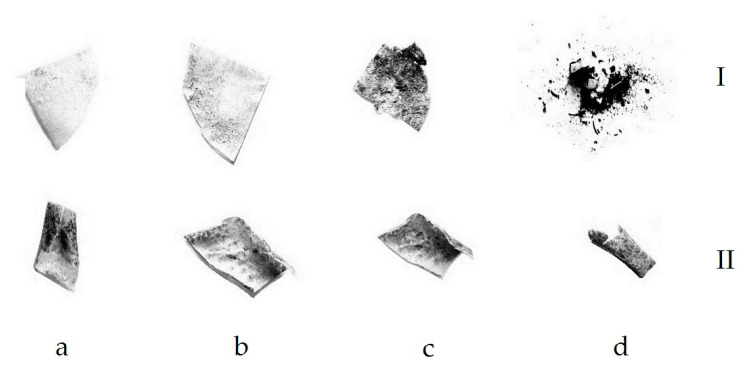
Photographs of PLA-chitosan films (I) and films containing sorbed iron ions (II), after exposition in soil during 2 (**a**), 4 (**b**), 7 (**c**) and 12 (**d**) weeks.

**Figure 8 polymers-15-00645-f008:**
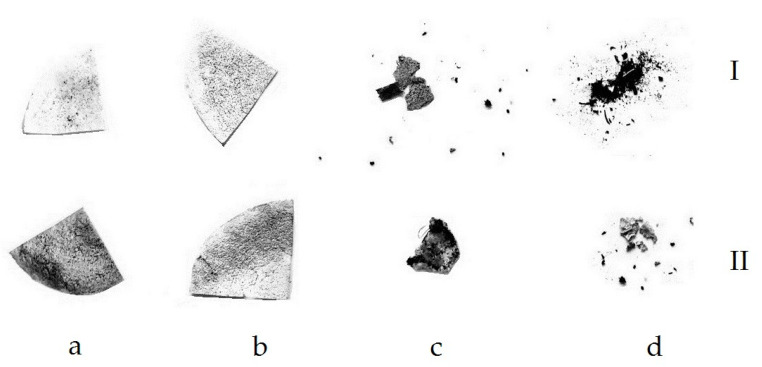
Photographs of PHB-chitosan films (I) and films containing sorbed iron ions (II), after exposition in soil during 2 (**a**), 4 (**b**), 7 (**c**) and 12 (**d**) weeks.

**Figure 9 polymers-15-00645-f009:**
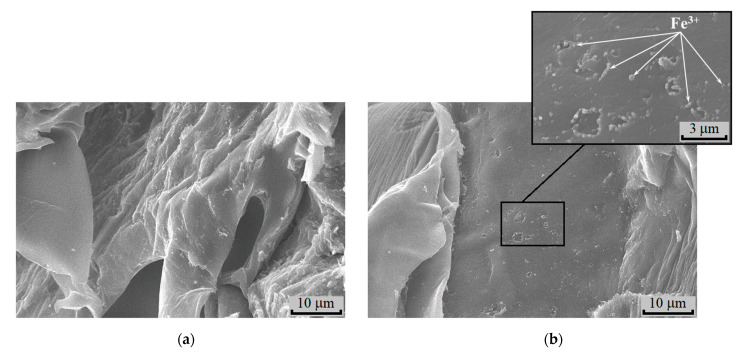
SEM micrographs of the initial films PLA-chitosan (50:50, wt. %) (**a**) and after sorption (**b**) from 0.3 mol/L FeCl_3_ solution.

**Figure 10 polymers-15-00645-f010:**
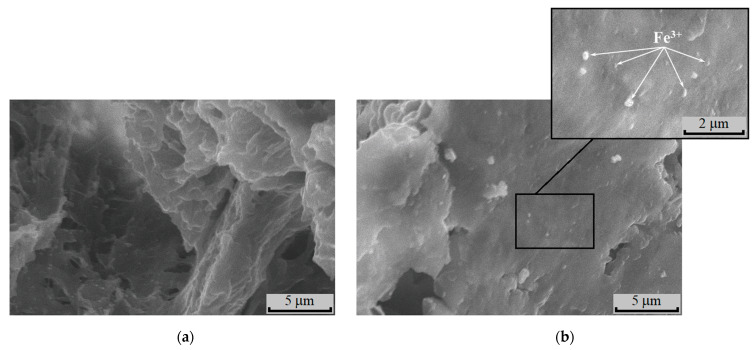
SEM micrographs of the initial films PHB-chitosan (50:50, wt. %) (**a**) and after sorption (**b**) from 0.3 mol/L FeCl_3_ solution.

**Table 1 polymers-15-00645-t001:** Parameters of Fe^3+^ sorption by binary composite films PLA-chitosan and PHB-chitosan (50:50 wt. %).

Cv(Fe^3+^), mol/L	k, h^−1^	Intercept of Equation (3)	R-sq. COP	Pierson’s r
PLA-chitosan				
0.002	0.513 ± 0.053	0.051 ± 0.010	0.979	0.989
0.005	0.168 ± 0.043	0.101 ± 0.051	0.997	0.998
0.008	0.070 ± 0.081	0.089 ± 0.009	0.962	0.981
PHB-chitosan				
0.002	0.267 ± 0.08	0.159 ± 0.032	0.998	0.993
0.005	0.129 ± 0.06	0.234 ± 0.063	0.992	0.996
0.008	0.109 ± 0.04	0.169 ± 0.036	0.996	0.998

## Data Availability

The data presented in this study are available on request from the corresponding author.

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
