# Peer review of "Hydrolysis, Biodegradation and Ion Sorption in Binary Biocomposites of Chitosan with Polyesters: Polylactide and Poly(3-Hydroxybutyrate)"

_polymers, 2023, doi:10.3390/polym15030645_

Round 1

Reviewer 1 Report

Here are some minor comments to improve the quality of the paper.

 ·        The manuscript needs to be reviewed one more time by the authors to check the typos in the paper such as "Henry's low"!! in the abstract.

·        The purpose of the work is mentioned in the last paragraph of the introduction, but you need to clarify the novelty of your work here.

·        Clarify the basis and reason you used these solution concentrations.

·        Why have you studied hydrolysis at high temperatures such as 70oC?

·        It would be better to present the main achievements of the work in bullets in the conclusion section.

·        The equations need to be referred properly.

·         Use the same word in the whole manuscript. Use either “sorption” or “absorption”.

·        The quality of Figures 7 and 8 is not suitable.

·        Page 10, line 312: “As can be seen from the figure, the sorption of iron ions leads to a change in the color of the samples.” How do you see their color change while in black and white?

·        Add a picture of your experiment work in the section “2. Materials and Methods”.

·        In figures 9 and 10, use arrows to show each image's materials/particle type.

Author Response

Dear reviewer!

Thank you for attentive reading of our manuscript and made comments. The necessary corrections in the text were made. The answers on comments are presented below.

Reviewer 2 Report

In this Manuscript entitled “Hydrolysis, Biodegradation and Ion Sorption in Binary Bio-composites of Chitosan with Polyethers Polylactide and Poly(3-Hydroxybutyrate)”, the authors fabricated chitosan/PLA and chitosan/PHB-based binary composites and explored their functional characteristics like hydrolysis resistance, biodegradation in soil, and ion sorption behavior.

Overall, the research article has some shortcomings, which need to be addressed before possible publication in polymers journal.

Please find the attached annotated file to see my comments.

Lastly, I would like to say “Polymers” journal publishes high-quality research articles related to biopolymers. Based on my comments, the recommendation is Major Revision.

Author Response

Dear reviewer!

Thank you for attentive reading of our manuscript and made comments. The necessary corrections in the text were made. The answers on comments are presented in PDF file.

Round 2

Reviewer 2 Report

Accept as it is